# *Lactobacilli* in COVID-19: A Systematic Review Based on Next-Generation Sequencing Studies

**DOI:** 10.3390/microorganisms12020284

**Published:** 2024-01-29

**Authors:** Clarissa Reginato Taufer, Pabulo Henrique Rampelotto

**Affiliations:** 1Graduate Program in Genetics and Molecular Biology, Universidade Federal do Rio Grande do Sul, Porto Alegre 91501-970, Brazil; 2Bioinformatics and Biostatistics Core Facility, Instituto de Ciências Básicas da Saúde, Universidade Federal do Rio Grande do Sul, Porto Alegre 91501-970, Brazil

**Keywords:** *Lactobacillus*, bacteria, microbiome, SARS-CoV-2, microbiota, host immune system

## Abstract

The global pandemic was caused by the SARS-CoV-2 virus, known as COVID-19, which primarily affects the respiratory and intestinal systems and impacts the microbial communities of patients. This systematic review involved a comprehensive search across the major literature databases to explore the relationship between lactobacilli and COVID-19. Our emphasis was on investigations employing NGS technologies to explore this connection. Our analysis of nine selected studies revealed that lactobacilli have a reduced abundance in the disease and an association with disease severity. The protective mechanisms of lactobacilli in COVID-19 and other viral infections are likely to be multifaceted, involving complex interactions between the microbiota, the host immune system, and the virus itself. Moreover, upon closely examining the NGS methodologies and associated statistical analyses in each research study, we have noted concerns regarding the approach used to delineate the varying abundance of lactobacilli, which involves potential biases and the exclusion of pertinent data elements. These findings provide new insight into the relationship between COVID-19 and lactobacilli, highlighting the potential for microbiota modulation in COVID-19 treatment.

## 1. Introduction

SARS-CoV-2 infection manifests with clinical symptoms affecting both the respiratory and gastrointestinal systems. The severity of these symptoms is associated with cytokine storms, where an excessive immune response can lead to severe inflammation and tissue damage, contributing to the development of more severe symptoms and complications [1]. In addition, evidence from several studies suggests that alterations in the gut microbiota also play a role in symptom development. Numerous studies underscore the significance of the human microbiota concerning both health and diseases, including COVID-19. Such studies have analyzed changes in the microbiota among COVID-19 patients to comprehend the connection between COVID-19 and the microbiota [2,3,4].

The assessment of the microbiota has revealed that COVID-19 patients’ microbiota is different compared to that of controls. Dysbiosis in patients is characterized by the depletion of beneficial commensals and the enrichment of opportunistic pathogens, resulting in significantly reduced bacterial diversity [5,6]. Previously, commensal bacteria that showed an inverse correlation with a SARS-CoV-2 load in fecal samples were identified as downregulating the expression of angiotensin-converting enzyme 2 (ACE2) in the murine gut [6,7]. The abundance of species also varies among patients exhibiting a signature of high or low-to-no SARS-CoV-2 infectivity in fecal samples [8]. *Collinsella aerofaciens*, *Collinsella tanakaei*, *Streptococcus infantis*, and *Morganella morganii* were increased in patients with a signature of high SARS-CoV-2 infectivity, while patients with a signature of low-to-no SARS-CoV-2 infectivity exhibited a higher abundance of *Parabacteroides merdae*, *Bacteroides stercoris*, *Alistipes onderdonkii*, and *Lachnospiraceae bacterium*. Additionally, changes in the community were associated with the severity of COVID-19, allowing discrimination between critical, general, and severe patients [9]. Lactobacilli, which are known for their diverse properties in human health and their significant variations in abundance under both healthy and diseased conditions, have been a focal point in these studies.

“Lactobacilli” is the generic term used to designate organisms in all 25 genera that are classified as *Lactobacillus* species. Recent advancements in microbial taxonomy have led to the reclassification of the genus *Lactobacillus*, resulting in the use of “lactobacilli” to collectively refer to multiple species within this genus This reclassification reflects a more comprehensive understanding of the genetic and phenotypic diversity among *Lactobacillus* species, prompting the adoption of a more inclusive and versatile nomenclature. 

Lactobacilli are gram-positive, non-spore-forming facultative or strict anaerobic bacteria. They are microorganisms adapted to their hosts, particularly in vertebrates, and have the capability to ferment a wide range of substrates. Despite representing <1% of the human intestinal bacterial population, their presence is constant in the intestinal microbiota [10]. The reduction of lactobacilli in the intestine is frequently linked to diseases, and the distinct species and specific strains of lactobacilli can contribute to variations in the taxa’s impact on diseases [11]. 

In recent years, next-generation sequencing (NGS) has emerged as the most suitable method for investigating microbial changes in a variety of human diseases, including viral infections [12]. Two primary approaches, namely amplicon sequencing and shotgun metagenomics, are employed to identify bacteria using NGS technologies. Amplicon sequencing involves the amplification and sequencing of the 16S ribosomal RNA (16S rRNA) gene by PCR [13]. This economical approach allows for focused analysis of microbial taxa in each sample, yielding valuable insights into the composition and diversity of the microbiota associated with various clinical conditions [14]. In contrast, shotgun metagenomics entails sequencing the complete microbial DNA without prior amplification or targeting, providing a broader perspective of the microbial community [15]. While shotgun metagenomics enables the discovery of novel pathogens and the exploration of functional aspects within microbial communities [16], it is more costly and computationally intensive compared to amplicon sequencing.

Since the onset of the COVID-19 pandemic, NGS technologies have played a pivotal role in investigating the correlation between bacterial communities and the disease [17]. These investigations have employed diverse sampling methodologies, sequencing tools, and statistical approaches. This diversity poses a challenge in synthesizing a comprehensive overview of the field and discerning the primary patterns in microbial composition associated with different stages of the disease. Notably, the examination of scientifically significant taxa, such as lactobacilli, adds complexity to understanding the intricate relationship between bacterial communities and COVID-19.

In this work, we examined the connection between lactobacilli and COVID-19, expanding its association with residents in the gut and URT (upper respiratory tract), which encompasses oral, nasal, and oropharyngeal sites. We conducted a systematic review to identify and summarize studies that demonstrated a relationship between the genus and the disease, exploring the underlying mechanisms involved. Our emphasis was on investigations employing NGS technologies to explore this connection, conducting a thorough analysis of the methodologies employed in addressing COVID-19 and unveiling the identification of this specific genus for the first time.

The underlying goal was to address a knowledge gap related to the gut microbiota in COVID-19 by identifying changes in a key microbial taxon (i.e., lactobacilli). By exploring this gap, the study aimed to contribute valuable insights into the mechanisms leading to microbial changes in COVID-19 and provide a foundation for future research.

## 2. Materials and Methods

### 2.1. Search Strategy and Selection Criteria

The systematic literature search was conducted using the Embase, Scopus, PubMed, and Web of Science databases. The search was carried out in August 2023 and focused on articles published from January 2020 to August 2023. The search strategy was developed using medical subject headings (MeSH) in the PubMed database and was adjusted for use in the other databases. Initially, the search was not restricted by language; however, only works published in English were included in this study. The detailed search strategy for each database is provided in Appendix A. As inclusion criteria, studies employing whole sequencing methodology or amplicon sequencing to evaluate the intestinal and upper respiratory tract microbiota in COVID-19 patients, both asymptomatic and symptomatic at different disease severities, were considered. Articles using methods other than sequencing and studies assessing the use of probiotics lactobacilli alone or in combination with other genera were excluded. Studies that did not meet these criteria were not included in the analysis. Additionally, review articles, book chapters, case reports, editorials, letters, notes, in vitro studies, and animal studies were excluded from consideration. Furthermore, articles cited by the studies identified in the searches were assessed, and those that aligned with the predetermined inclusion criteria were added to the analysis.

The resulting articles were first assessed using Rayyan software (http://rayyan.qcri.org, accessed on 24 January 2024) to identify and remove duplicates [18]. Subsequently, they were screened based on their abstracts to determine if they met the inclusion and exclusion criteria. Articles that satisfied the inclusion criteria underwent a full-text evaluation, and a data extraction was performed. Figure 1 illustrates the search process conducted following the PRISMA guidelines (Appendix A) [19].

### 2.2. Data Extraction and Analysis

Information from each paper was compiled in an Excel spreadsheet, encompassing general details about the work, study population details, methodology, and key findings (Appendix A).

To evaluate the studies’ quality, the critical appraisal tools from the Joanna Briggs Institute for case–control, cohort, and cross-sectional studies were used [20].

## 3. Results

### 3.1. Selected Studies

Table 1 displays the nine articles that fulfilled the inclusion criteria. These studies were carried out in various countries and encompassed diverse patient populations and disease severities. China had the highest number of papers with three, followed by Italy and the USA with two each, and Japan and Turkey with one each. The searches were not limited by language, and only four returned articles were in another language (Russian and Ukrainian). Nevertheless, we chose to include only works published in the English language. Four studies assessed the upper respiratory tract [21,22,23,24], including samples such as saliva, throat, tongue, oral, and nasopharyngeal. Four studies examined stool samples [25,26,27,28], and only one study evaluated the microbiome in both the gut and URT [29].

The majority of the studies used the 16S rRNA gene amplicon sequencing method to detect alterations in the gut microbiota of COVID-19 patients. One study focused on the microbiota of children [27]. The V3–V4 region was the most commonly used amplicon sequencing approach [21,25,28,29], while the V1–V2 region was used in two studies [21,25], and the V4 region was applied in one study [23] (Figure 2). Whole genome sequencing methodology was applied in two studies [20,24]. There was a notable variation in the selection of databases used for taxonomy classification among studies, with four studies using the Greengenes database [21,22,23,25]; the RDP [22], SILVA v132 [23], SILVA v138 [21], and MetaPhlAn2 (V.20) [28] databases were used in only one study each (Figure 2). One study did not specify the database [24].

In terms of the statistical analysis employed to detect differences in the amount of the lactobacilli between groups, LEfSe (Linear Discriminant Analysis Effect Size) was the most commonly used approach [21,25,26,27,28,29], while two studies used DESeq2 [22,23], and one study used the Kruskal–Wallis method [24] (Figure 2).

### 3.2. Risk of Bias and Quality Assessment

The risks of bias are depicted in the traffic light graphs in Figure 3. In the case–control studies, the most significant limitation identified was the failure to define criteria for identifying cases and controls, measure exposure consistently in both groups and, consequently, identify and address confounding factors and strategies to manage them. The study conducted by Gaibani et al. (2021) [25] used control samples from publicly available databases, introducing a substantial bias in the sample analysis. For the cohort studies, two studies failed to identify confounding factors, leading to a lack of strategies to address the confounding factors. Additionally, question number 6 (D6, regarding the initial assessment of the outcome) does not align, compromising the assessment of the risk of bias.

## 4. Discussion

### 4.1. The Lactobacilli

In 1901, Beijerinck introduced the genus *Lactobacillus*, initially relying on phenotypic characteristics, sugar utilization, optimal growth temperature, and the range of produced metabolites for its classification. As molecular identification techniques advanced, new criteria were established for classifying the genus. Consequently, the genetic diversity within the genus *Lactobacillus* was found to be higher than that typically observed in bacterial genera and families [29]. More recently, the family Lactobacillaceae and the genus *Lactobacillus* underwent a restructuring, resulting in the classification of 25 genera from *Lactobacillus*, 23 of which are new. Furthermore, a new description for the genus was proposed [30].

According to the new description, lactobacilli species are gram-positive, homo-fermentative, and non-sporulating rods [29]. These species are host-adapted, with the *Lactobacillus melliventris* clade being adapted to social bees, while all the others are adapted to vertebrates. They can ferment a wide range of carbohydrates and exhibit strain-specific capabilities to ferment extracellular fructans, starch, or glycogen. Many species can ferment mannitol and do not ferment pentoses [31,32]. 

Since the taxonomic restructuring of *Lactobacillus* is recent, the reference databases used for taxonomic classification in NGS studies are still incomplete. Therefore, we have retained the term “Lactobacillus” in reference to the results of these selected studies in this review, while using “lactobacilli” to refer to the present context. 

### 4.2. Evidence in COVID-19

In the oral samples, the relative abundance of lactobacilli increased in the COVID-19 patients compared to the controls. At the species level, increased quantities of *Lactobacillus fermentum* were identified [24]. 

The oral microbiota was also assessed in two different patient groups (mild–moderate, severe–critical) and a control group. In the severe–critical patients, the LEfSe analysis showed a higher abundance of lactobacilli, making it one of the top five bacterial biomarker genera for the group. This suggests a potential association between the oral microbiome and the severity in patients [21].

Differences in the microbiota were also assessed in Intensive Care Unit (ICU) and non-ICU patients through differential abundance analysis (DESeq2). Additionally, for the oral microbiota (saliva samples), Kim et al. demonstrated that lactobacilli were significantly more abundant in the non-ICU group, while for nasopharyngeal samples, the genus was significantly more abundant in the ICU group [22]. On the other hand, using the combined approach of LEfSe and MaAsLin2, Wu and colleagues did not identify the genus as significantly altered in oral samples [29].

For the evaluation of the intestinal microbiota, lactobacilli was one of the taxa increased in COVID-19 patients, determined significantly different by LEfSe [29]; furthermore, it was identified as one of the three characteristic genera of the severe group by the LEfSe. The relative abundance was not different between severe patient groups who received or did not receive antibiotic treatment [26].

The enrichment of the genus in the gut microbiota of patients was also demonstrated by Gaibani et al. [25]. At the species level, the differential abundance of a lactobacilli ASV (identified as *Lactobacillus jensenii* by the BLAST database) was decreased at all seven time points assessed in a longitudinal study of upper respiratory tract samples (days 1, 3, 5, 7, 10, 14, and 21) [23]. In children, the LEfSe assessment of the intestinal microbiota showed that *Lactobacillus ruminis* was one of the dominant species in the control group compared to COVID-19 patients and those with both COVID-19 and multisystem inflammatory syndrome (MIS-C) [27]. The intestinal microbiota of ten recovered severe patients was also evaluated, and it was observed that the relative abundance of lactobacilli was dominant in some of the recovered patients [26]. At the species level, *L. ruminis* was identified as enriched (LEfSe) in samples collected up to 30 days after they tested negative for SARS-CoV-2, regardless of antibiotic treatment [28].

In summary, numerous studies have explored the role of lactobacilli in COVID-19. In oral samples, the relative abundance of lactobacilli was observed to be higher in COVID-19 patients compared to controls, with *L. fermentum* identified as one of the species with increased quantities. The abundance of lactobacilli in the oral microbiota was also found to be associated with disease severity, with severe–critical COVID-19 patients showing a higher abundance of lactobacilli. Additionally, the abundance of lactobacilli in oral samples was significantly higher in non-ICU patients, while in nasopharyngeal samples, it was significantly higher in ICU patients. In the intestinal microbiota, *Lactobacillus* was increased in COVID-19 patients and identified as one of the characteristic genera in severe patient groups. A specific lactobacilli species (*L. jensenii*) showed decreased abundance over time in a longitudinal study of upper respiratory tract samples. However, *L. ruminis* was one of the dominant species in the intestinal microbiota of the control group compared to COVID-19 patients and those with MIS-C. The relative abundance of *Lactobacillus* was dominant in some recovered severe patients, and *L. ruminis* was enriched in samples collected up to 30 days after patients tested negative for SARS-CoV-2, regardless of antibiotic treatment. These findings suggest that lactobacilli may play a role in COVID-19, particularly in relation to oral and intestinal microbiota composition and disease severity. Further research is needed to fully understand the implications and potential therapeutic applications. 

### 4.3. Risk of Bias and Quality Assessment

The analysis of bias risk revealed that the primary bias in case–control and cohort studies was the inconsistent measurement of exposure, leading to difficulties in identifying and addressing confounding factors and implementing strategies to manage them. In case-control studies, there was also a lack of information regarding the selection of control patients. These information gaps about the groups and the failure to identify and consider confounding factors can result in biased findings and imprecise deductions, potentially impacting the reliability and validity of the study’s conclusions. Observational studies carry a notably high risk of confounding factors due to their intrinsic nature. The failure to identify and consider confounding factors in these studies can result in biased findings. As a result, it is essential for researchers to thoughtfully assess potential confounding variables and utilize suitable statistical methods to reduce the impact of confounding factors. 

The overview of microbiome methods employed to detect and measure lactobacilli also provides essential insights into the study’s quality. Most studies that focused on amplicon sequencing utilized the standard V3–V4 region commonly employed in microbiome studies. On the other hand, only one study used shotgun metagenomics for sequencing. Concerns arise when examining the databases and statistical methods used. There is a wide variety of databases and statistical methods among the studies, making direct comparisons challenging, as these variables can significantly influence taxon identification or abundance quantification. Additionally, many studies employed unconventional and less appropriate approaches. For example, the Greengenes database is outdated, and although the NCBI database is extensive, it does not offer proper alignment for taxonomy inference. In terms of statistical analysis, many studies employed suitable microbiome methods such as LEfSe, while others relied on simpler statistical approaches like Kruskal–Wallis. In addition to these concerns, the use of different primers presents challenges for gaining a comprehensive understanding of the relationship between lactobacilli and COVID-19, potentially compromising result consistency. 

These variations in microbiome methods, databases, and statistical approaches among studies can make direct comparisons challenging. The observed variability may significantly influence taxon identification and abundance quantification, impacting the overall quality and reliability of the conclusions.

Therefore, it is highly advisable to implement reporting guidelines in human microbiota research, like the STORMS checklist [33]. Adhering to standardized guidelines enhances the transparency and quality of studies, facilitating more reliable conclusions. 

### 4.4. Evidence in Other Viral Infections

The importance of lactobacilli for health and control of viral diseases has been studied mainly by evaluating the administration of lactobacilli probiotics.

In adults, consuming *Lactobacillus plantarum* HEAL 9 (DSM 15312) and *Lactobacillus paracasei* 8700:2 (DSM 13434) reduces the incidence of one or more episodes of the common cold, the number of days that symptoms persist, and the severity of pharyngeal symptoms in healthy adults. When compared to the control group, the probiotics considerably inhibited the growth of B lymphocytes [34]. When given to healthy people following seasonal influenza vaccination, *Lactobacillus casei* 431 probiotics also improved the immune response [35]. The probiotic-treated participants showed noticeably greater levels of vaccine-specific plasma IgG, IgG1, and IgG3. In addition, oral *Lactobacillus* fermentum VRI 003 treatment was linked to a significant decrease in the duration and intensity of respiratory disease in elite athletes [36].

In children between the ages of three and six, consuming bulgaricus-fermented dairy products decreased the incidence rate of common infectious illnesses by 19%. Specifically, the incidence rate of digestive infections was 24% lower in the treated group than in the control group. Furthermore, compared to the control group, the active group showed an 18% reduced incidence rate of infections of the upper respiratory system. Finally, compared to the control group, the occurrence of infections of the lower respiratory system decreased by 2% in the active group [37]. Giving *Lactobacillus* GG as a probiotic has also been linked to a markedly lower incidence of acute infectious diarrhea in kids and babies.

In the elderly, the consumption of yogurt fermented with *Lactobacillus delbrueckii* ssp. *bulgaricus* OLL1073R-1 among healthy elderly people showed a 2.3-fold decrease in the risk of getting the common cold when compared to the group that only drank milk [38].

A growing body of evidence from experimental studies also indicates that lactobacilli may play a crucial role in modulating the immune response and influencing viral outcomes. 

In mice, heat-killed *L. casei* DK128 protected against H3N2 virus infections [39], and oral administration of heat-killed *L. plantarum* L-137 reduced viral titers after influenza virus infection [40]. Yogurt fermented with *L. delbrueckii* ssp. *bulgaricus* OLL1073R-1, along with its exopolysaccharides, demonstrated anti-influenza virus effects in mice [41]. *Lactobacillus pentosus* b240 and *L. plantarum* DK119 extended the survival period and reduced viral titers in mice infected with the influenza virus [42,43]. Mice were protected against influenza virus infection through the intranasal administration of live lactobacilli species [44]. Additionally, intranasal inoculation of live *L. plantarum* and *Lactobacillus reuteri* protected against pneumovirus infection [45].

In summary, these findings suggest that various strains of lactobacilli may have potential benefits in preventing and reducing viral infections, particularly in the respiratory and gastrointestinal tracts.

### 4.5. Mechanisms of Protection of Lactobacilli in COVID-19

While the exact protective mechanisms of lactobacilli against COVID-19 and other viral infections are not fully understood, several studies suggest potential avenues. How lactobacilli can direct antiviral effects, lower inflammation, and regulate the immune system are covered in this section.

#### 4.5.1. Direct Antiviral Effects

Insights into the direct antiviral mechanisms of lactobacilli in the context of COVID-19 encompass various strategies:

Production of Antiviral Substances: Certain actobacilli species produce antimicrobial peptides and substances inhibiting viral entry and replication [46].

Competition for Cellular Binding Sites: Lactobacilli compete with pathogenic viruses for binding sites, limiting viral attachment and entry. Some strains produce bacteriocins and exopolysaccharides inhibiting viral replication [47].

Receptor Binding and Upregulation: Certain lactobacilli strains bind to enterocyte receptors, stimulating the upregulation of MUC-2 and MUC-3, inhibiting bacterial translocation and pathogen adhesion [48,49].

Release of ACE-Inhibiting Peptides: Lactobacilli release peptides inhibiting ACE, potentially preventing virus entry. Probiotics, especially lactic acid bacteria, produce ACE-inhibiting peptides [50].

Computational Support: Computational studies on metabolites in *L. plantarum* demonstrate their effectiveness in blocking viral entry by binding to RdRp, RBD, and ACE2 [51].

Direct Interference in Viral Replication: Lactococcin G in *L. plantarum* interferes with viral replication by binding to viral proteins, suggesting the broader potential of bio-antimicrobial peptides for COVID-19 [52].

Antiviral Activity of *L. plantarum* Probio-88: Molecular docking supports the potential adjuvant therapeutic effect of *L. plantarum* Probio-88 in viral infections [53].

#### 4.5.2. Immunomodulatory Effects

Several studies highlight the immunomodulatory effects of lactobacilli strains on the immune response, revealing diverse mechanisms:

NK Cell Stimulation: Lactobacilli stimulates natural killer (NK) cells, crucial for early defense against viral infections [54].

TLR2 and Immunoregulatory Molecules: Different strains influence Toll-like receptor 2 (TLR2), inducing heightened TLR2 expression and stimulating the production of immunoregulatory molecules, including IL-10, IFN-γ, and TNF-α [55,56].

TLR4 Interaction: Various lactobacilli species engage with Toll-like receptor 4 (TLR4), affecting immune responses. For example, *L. casei* enhances IL-10, IFN-γ, and IL-6 production while decreasing TNF-α levels through TLR4 interaction [57].

NOD-Like Receptors (NLRs): Many lactobacilli species modulate immune responses through NLRs. *L. gasseri* and *L. delbrueckii* upregulate NLRP3 expression, ensuring proper NLRP3 activation [58]. *Lactobacillus salivarius* induces an anti-inflammatory effect by regulating NOD2 and promoting IL-10 production [59].

Broad Immunomodulation: Lactic acid bacteria, including lactobacilli, show potential to enhance human and animal health by modulating mucosal and systemic immune responses. They can suppress viral proliferation and protect against various viruses [60,61].

Gut–Lung Axis: Studies on the gut–lung axis emphasize lactobacilli’s ability to interact with immune cells, influencing cytokine production. This suggests a potential role in shaping the immune response to infections, including respiratory viruses like SARS-CoV-2 [62].

Metabolites and Immunomodulation: Lactobacilli may contribute to immune modulation by producing short-chain fatty acids (SCFA) and other metabolites with immunomodulatory effects [63].

#### 4.5.3. Anti-Inflammatory Effects

Lactobacilli demonstrate a dual impact on cytokine modulation, favoring anti-inflammatory responses and inhibiting pro-inflammatory signals. Different strains exhibit specific effects:

Cytokine Modulation: Various lactobacilli strains enhance antiviral cytokines (IL-12, IFN-γ) while reducing inflammatory ones (IL-4, IL-6, TNF-α) [41,43].

TNF-α Inhibition: Lactobacilli mitigate TNF-α-induced IL-8 production [64].

Butyric Acid Production: Mycelium fermentation of *Lactobacillus rhamnosus* EH8 produces butyric acid, downregulating PDE4B expression and IL-6 secretion in macrophages [65].

NF-κB Pathway Regulation: The NF-κB pathway is a key mechanism in cytokine modulation. Different lactobacilli strains actively modulate NF-κB activation:*L. casei* inhibits NF-κB pathway activation induced by *Shigella flexneri* [66].*L. rhamnosus* and *Lactobacillus helveticus* downregulate Th1 pro-inflammatory responses and enhance Th2 responses during *Citrobacter rodentium* infection [67].*L. reuteri*, *L. casei*, and *L. paracasei* exhibit anti-inflammatory properties by modulating NF-κB pathway elements, enhancing anti-inflammatory cytokine production and promoting cell survival [68,69,70,71].*plantarum* reduces NF-κB-activating factors by diminishing NF-κB-binding activity [72].*Lactobacillus brevis* prevents IRAK1 and AKT phosphorylation [73].

### 4.6. Comparison between Lactobacilli and Bifidobacterium in COVID-19

In this study, we conducted a systematic review to investigate the role of lactobacilli in COVID-19 using next-generation sequencing approaches. Our analysis identified nine original studies that specifically explored the presence and abundance of lactobacilli in COVID-19 patients. In comparison to our previous work focused on *Bifidobacterium*, which identified eighteen articles highlighting the differential abundance of this genus in COVID-19 patients, only three articles identified both lactobacilli and *Bifidobacterium* in the same study [21,26,27].

Lactobacilli and *Bifidobacterium* are both probiotic bacteria that have been studied for their potential immunomodulatory effects [63,74]. However, they may have distinct molecular mechanisms that could affect their involvement in COVID-19. Lactobacilli species are known to produce various metabolites, such as short-chain fatty acids, which can modulate immune responses and promote anti-inflammatory effects [63]. These SCFAs can influence immune cell function, cytokine production, and barrier integrity in the respiratory tract. Moreover, certain strains of lactobacilli have demonstrated the ability to produce antiviral compounds, such as bacteriocins and exopolysaccharides [47]. These substances exhibit the capability to impede viral replication and provide protection against viral infections. Studies focusing on lactobacilli in COVID-19 might explore these specific mechanisms and their impact on disease outcomes. 

On the other hand, *Bifidobacterium* species have been associated with the production of certain molecules like extracellular vesicles or specific metabolites like acetate and lactate [75]. Extracellular vesicles released by *Bifidobacterium* have been shown to have immunomodulatory properties, including anti-inflammatory effects. In addition, *Bifidobacterium* exhibits a competitive advantage over pathogenic bacteria, which can effectively hinder the colonization of other pathogens, potentially preventing gut dysbiosis [76,77,78]. It is worth noting that gut dysbiosis has been linked to the severity of COVID-19. These unique molecular mechanisms might be investigated in studies focusing on the role of *Bifidobacterium* in COVID-19.

Considering the differences in molecular mechanisms and their potential impact on COVID-19, it is understandable that there is limited overlap of articles between the two systematic reviews.

The recognition of distinct mechanisms employed by lactobacilli and *Bifidobacterium* in the context of COVID-19 raises intriguing possibilities for understanding their specific roles and clinical implications. The differences suggest the need for targeted studies exploring the specific impacts of lactobacilli and *Bifidobacterium* on COVID-19 outcomes. Investigating the specific roles of lactobacilli and *Bifidobacterium* can inform targeted interventions, offering a nuanced approach to managing COVID-19. By recognizing the unique strengths of each probiotic, researchers and clinicians may unlock new avenues for therapeutic strategies, aiming to harness the potential of the gut microbiota in mitigating the impact of SARS-CoV-2.

## 5. Conclusions

In conclusion, the studies identified in this systematic review suggest that lactobacilli may play a role in COVID-19, particularly in relation to oral and intestinal microbiota composition and disease severity. Furthermore, evidence from studies on other viral infections suggests that various strains of lactobacilli may have potential benefits in preventing and reducing viral infections, particularly in the gut and URT. The protective mechanisms of lactobacilli in COVID-19 and other viral infections are expected to be multi-faceted, involving intricate interactions among the microbiome, the host immune system, and the virus. This combination of protective mechanisms, including direct antiviral effects, immunomodulation, and enhancement of mucosal immunity, makes this genus a promising avenue for interventions against viral infections, including COVID-19.

## Figures and Tables

**Figure 1 microorganisms-12-00284-f001:**
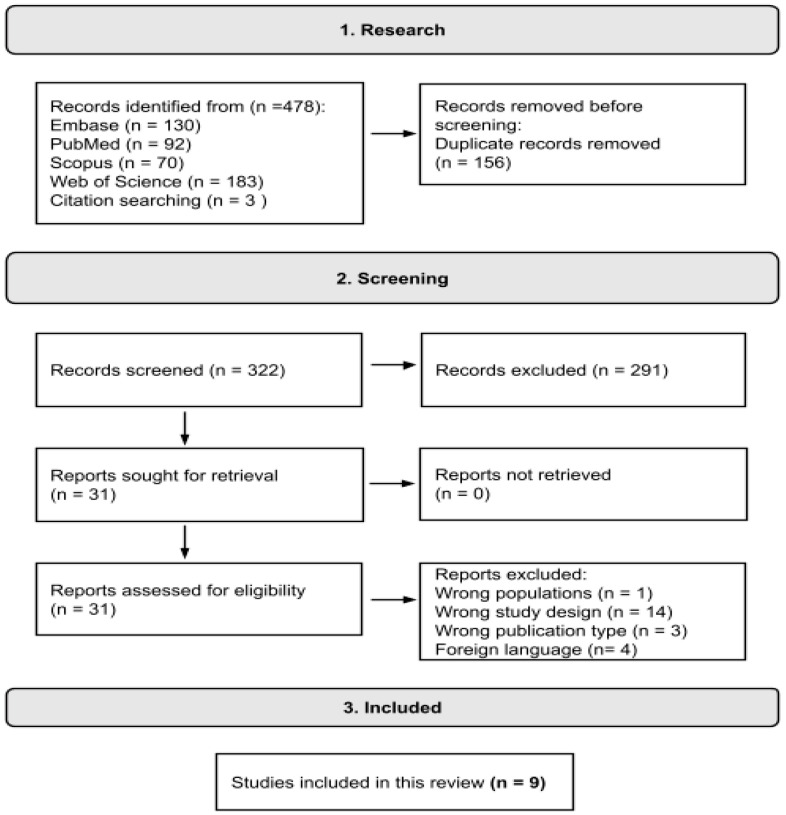
Diagram illustrating the process of searching the database.

**Figure 2 microorganisms-12-00284-f002:**
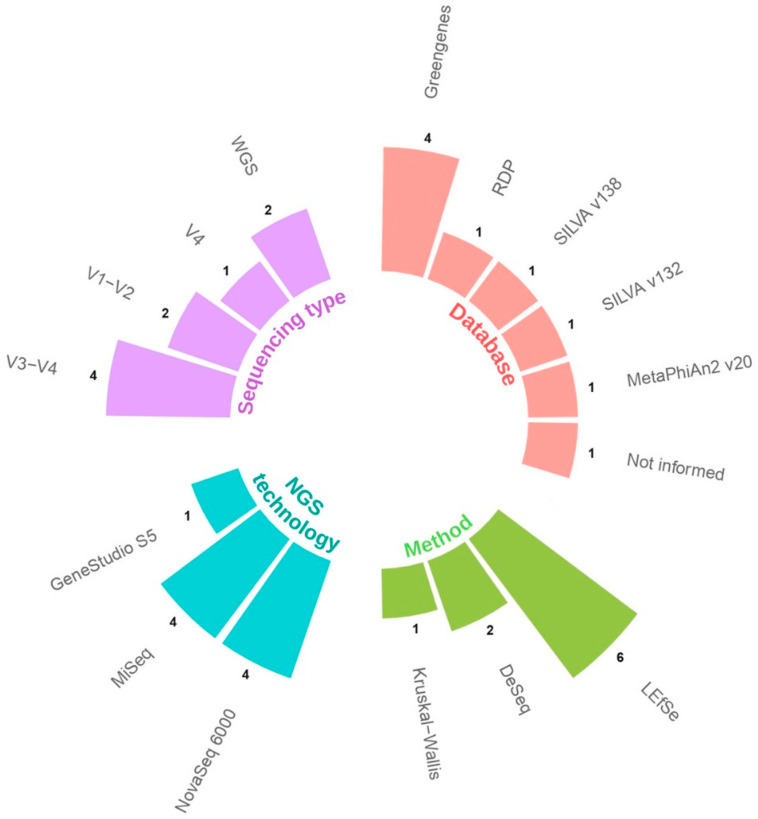
Bar graph representing the NGS technology, sequencing type, amplicon regions, taxonomy databases used, and statistical analysis methods employed to identify and measure lactobacilli.

**Figure 3 microorganisms-12-00284-f003:**
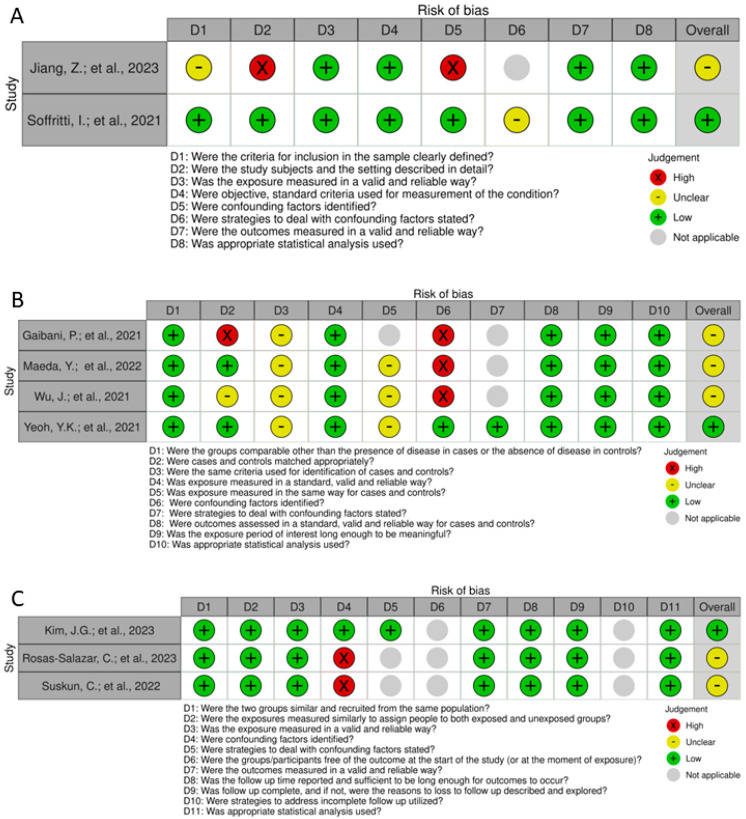
Traffic lights used for assessing the risk of bias in cross-sectional studies [20,23] (**A**), case—control studies [24,25,27,28] (**B**), and cohort studies [21,22,26] (**C**).

**Table 1 microorganisms-12-00284-t001:** Details of the papers identified through the systematic search.

Study	Country	Type of Study	Type of Sample	NGS Technology	Type of Sequencing	N	Groups	Abundance in COVID-19 Group	Clinical Relevance
Gaibani, P., et al., 2021 [24]	Italy	Case–control	Stool	MiSeq	V3–V4	138	69 COVID-19 patients 69 healthy controls ^1^	Enriched	Associated with disease
Jiang, Z., et al., 2023 [20]	China	Cross-sectional	Tongue/Oral	Novaseq 6000	V3–V4	130	49 mild to moderate COVID-19 44 severe and critical COVID-19 37 healthy control	Greater abundance in severe–critical patients. Detected as a group biomarker	Associated with severity
Kim, J.G., et al., 2023 [21]	USA	Cohort	Saliva and nasopharyngeal	MiSeq	V1–V2	NI	114 samples COVID-19 positive 30 samples COVID-19 negative	Depleted in oral samples and increased in nasopharyngeal samples from ICU patients	Associated with severity
Maeda, Y., et al., 2022 [25]	Japan	Case–control	Stool	MiSeq	V1–V2	108	40 severe COVID-19 38 mild COVID-19 30 healthy control	Characteristic of the severe group. Dominant relative abundance in some recovered patients.	Associated with disease severity and recovery process
Rosas-Salazar, C., et al., 2023 [22]	USA	Cohort	Upper Respiratory Tract	MiSeq	V4	48	24 mild-to-moderate COVID-19 patients 24 asymptomatic uninfected control	Differentially abundant. Decreased *Lactobacillus jensenii* in COVID-19.	Associated with disease
Soffritti, I., et al., 2021 [23]	Italy	Cross-sectional	Oral	Ion Gene Studio S5	Shotgun	75	39 COVID-19 patients 36 controls	Increase relative abundance. Increase *Lactobacillus fermentum*.	Associated with disease
Suskun, C., et al., 2022 [26]	Turkey	Cohort	Stool	NovaSeq 6000	V3–V4	39	20 COVID-19 patients 19 healthy controls	*Lactobacillus ruminis* dominating the control group.	Not associated with disease
Wu, Y.J., et al., 2021 [28]	China	Case–control	Throat and stool	NovaSeq 6000	V3–V4	129	53 COVID-19 patients 76 healthy controls	Increased in fecal samples.	Associated with disease
Yeoh, Y.K., et al., 2021 [27]	China	Case–control	Stool	NovaSeq 6000	Shotgun	178	100 COVID-19 patients 78 non-COVID-19 patients	Enriched *Lactobacillus ruminis* in recovered patients.	Associated with recovery process

^1^ From databank.

## Data Availability

Not applicable.

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
