# Peer review of "Lactobacilli* in COVID-19: A Systematic Review Based on Next-Generation Sequencing Studies"

_microorganisms, 2024, doi:10.3390/microorganisms12020284_

Round 1

Reviewer 1 Report

Comments and Suggestions for Authors

The article titled "Lactobacilli in COVID-19: A Systematic Review Based on Next-Generation Sequencing Study" is a comprehensive systematic review exploring the relationship between lactobacilli and COVID-19. The authors focus on research using Next-Generation Sequencing (NGS) technology to understand this connection. The review analyzes nine selected studies, finding a reduced abundance of lactobacilli in COVID-19 cases and a correlation with the severity of the disease. The paper discusses potential protective mechanisms of lactobacilli in COVID-19 and other viral infections, which may be complex and involve interactions between the microbiome, host immune system, and virus. Concerns about the methods used in these studies, including potential biases and exclusion of relevant data, are also discussed. These findings provide new insights into the relationship between COVID-19 and lactobacilli, emphasizing the potential for microbiome modulation in COVID-19 treatment.

    1. Selecting only nine studies might limit the scope and universality of the findings. A broader range of studies could provide a more comprehensive understanding.   2. This review focuses on studies using Next-Generation Sequencing, potentially overlooking insights from other research methodologies.   3. Although the review indicates potential for microbiome modulation in COVID-19 treatment, more direct evidence or clinical trial data would strengthen this claim.

Comments on the Quality of English Language

no

Author Response

We thank the reviewer for the comments and suggestions. While including only nine studies may limit the scope of the findings, it's important to note that we aimed to focus on high-quality research using Next-Generation Sequencing (NGS) technology to ensure a thorough and detailed analysis of this emerging technology. Additionally, these studies combine a total of 989 samples, which is considerable for NGS evaluation. More direct evidence of the role of Lactobacillus in COVID-19 treatment is presented in a complementary systematic review focused on the impact of probiotic Lactobacillus on COVID-19. Given the different nature and methods of both studies (evaluating the impact of probiotics is completely different than analyzing the abundance of microbial taxa through NGS), they had to be performed separately.

Reviewer 2 Report

Comments and Suggestions for Authors

The text addresses the relationship between the presence of the Lactobacillus bacteria genus and the COVID-19 disease, exploring studies that use next-generation sequencing to understand this connection. The studies analyze changes in the intestinal and upper respiratory tract microbiota in patients with COVID-19, at different stages of the disease. However, the text highlights some limitations in the studies reviewed, such as the lack of consistent criteria for identifying cases and controls, in addition to the diversity of methods used for genetic analysis, which makes it difficult to directly compare results between studies. Furthermore, there are other problems. In the introduction, authors should include recent studies' citations to enhance the relevance of the background. Expand on how microbiota changes might influence disease severity beyond Lactobacillus. In methods, clarify the rationale for exclusion criteria, especially studies examining probiotics with Lactobacillus. Describe the rationale behind selecting specific databases and taxonomic classifications. In the discussion, there are Limited discussion on potential biases in selected studies. therefore, authors should discuss potential biases or limitations within individual studies, addressing heterogeneity in methodologies, sample sizes, or patient populations. Overwhelming focus on animal models; consider balancing with human studies. Therefore, authors should Include more human-based studies or clinical trials showcasing Lactobacillus's effects on viral infections. Incorporate recent studies or clinical evidence supporting these mechanisms in COVID-19 cases, this is the heart of the manuscript. There is limited in-depth analysis of their molecular mechanisms and potential effects on COVID-19 outcomes. Therefore, it would be crucial to explore specific molecular pathways or immune interactions that distinguish these genes in relevance to COVID-19.

Author Response

We thank the reviewer for the comments and suggestions. We have considered them all and have revised the manuscript accordingly. Concerning the rationale for exclusion criteria, we specifically focused on studies that used next-generation sequencing (NGS) technologies to provide, for the first time, an in-depth analysis of these emerging methods and the identification of this particular genus. Regarding the exclusion of probiotic studies, evaluating the impact of probiotics is completely different than analyzing the abundance of microbial taxa through NGS. Given the different nature and methods of both studies, they had to be performed separately. For this reason, the impact of probiotic Lactobacillus on COVID-19 is presented in a complementary systematic review. The rationale behind selecting specific databases and taxonomic classifications is crucial for ensuring the accuracy and relevance of the results. By carefully choosing databases and taxonomic classifications, we aim to capture a comprehensive and representative range of data related to lactobacilli and COVID-19. This approach allows us to focus on specific information that is most pertinent to our research question, ensuring a thorough and targeted analysis of the relationship between lactobacilli and COVID-19. We have also reorganized and included more references in Section 4.4 for better-balancing human and animal studies. In addition, we have also expanded and reorganized Section 4.5 to better explore specific molecular pathways.

Reviewer 3 Report

Comments and Suggestions for Authors

In this paper authors explored the literature databases to find a relationship between Lactobacillus and COVID-19 using NGS technologies to explore this connection. This subject is important taking into account recent pandemic situation. The abstract o this paper is clear and concise. In introduction authors present the situation related to COVID-19 and Lactobacillus by using NGS technologies which played a pivotal role in investigating the correlation between bacterial communities and the disease. A systematic literature search was conducted using Embase, Scopus, PubMed, and Web of Science databases. The critical appraisal tools from the Joanna Briggs  Institute for case-control, cohort, and cross-sectional studies were used and nine papers fulfilled all criteria. Based on these papers authors concluded that Lactobacillus may play a role in COVID-19, particularly concerning oral and intestinal microbiota composition and disease severity. The protective mechanisms in COVID-19 and other viral infections are expected to be multi-faceted, involving intricate interactions among the microbiome, the host immune system, and the virus.

The paper is well-written and with important data for researchers in this field.

Author Response

We thank the reviewer for the comments.

Round 2

Reviewer 2 Report

Comments and Suggestions for Authors

The Manuscript underwent major transformation and many sections were substantially altered. It is undeniable that there have been improvements, however, there are still points that deserve attention:

In the introduction, the authors could highlight more emphatically the specific knowledge gap that the study aims to fill and

It would be useful to specify the role of "cytokine storms" in the symptoms of COVID-19, mentioned in the opening paragraph. Regarding the methodology, it would be interesting if the authors could provide a brief justification for the choice of databases and sequencing regions chosen. In the results and discussion, did the aurors think about including information about the distribution of patients into different disease severity groups? More information about the differences in outcomes and clinical implications between Lactobacillus and Bifidobacterium would be valuable. Explain how high risk of bias may affect conclusions.

Author Response

We thank the reviewer for the additional comments. We have provided below a point-by-point reply to them all. All alterations in the revised manuscript were highlighted in red.

In the introduction, the authors could highlight more emphatically the specific knowledge gap that the study aims to fill

Reply: We addressed this comment in lines 89-93.

and it would be useful to specify the role of "cytokine storms" in the symptoms of COVID-19, mentioned in the opening paragraph.

Reply: We addressed this comment in lines 29-30.

Regarding the methodology, it would be interesting if the authors could provide a brief justification for the choice of databases and sequencing regions chosen.

Reply: I am not sure if I understood this comment quite well because the methodology does not involve the choice of databases and sequencing regions. That’s part of what was found in the selected articles, so they were described in the results (lines 149-158) and properly discussed (274-286).

In the results and discussion, did the authors think about including information about the distribution of patients into different disease severity groups?

Reply: I believe we have already addressed this issue when specifying the different disease severity groups in Table 1 and properly discussing each article based on the severity groups.

More information about the differences in outcomes and clinical implications between Lactobacillus and Bifidobacterium would be valuable.

Reply: We addressed this comment in lines 456-464.

Explain how high risk of bias may affect conclusions.

Reply: We have addressed this issue throughout Section 4.3 (see red texts added)